# Advanced Pediatric Diffuse Pontine Glioma Murine Models Pave the Way towards Precision Medicine

**DOI:** 10.3390/cancers13051114

**Published:** 2021-03-05

**Authors:** Zirong Chen, Peng Peng, Xiaolin Zhang, Barbara Mania-Farnell, Guifa Xi, Feng Wan

**Affiliations:** 1Department of Neurological Surgery, Tongji Hospital, Tongji Medical College, Huazhong University Science and Technology, Wuhan 430030, China; czrong@hust.edu.cn (Z.C.); drpengpeng@hust.edu.cn (P.P.); zhangxl96718@hust.edu.cn (X.Z.); 2Department of Biological Science, Purdue University Northwest, Hammond, IN 46323, USA; bmania@pnw.edu; 3Department of Neurological Surgery, Northwestern University Feinberg School of Medicine, Chicago, IL 60611, USA

**Keywords:** diffuse intrinsic pontine glioma, molecular biology, patient derived xenografts, genetically engineered mouse model, humanized mouse model

## Abstract

**Simple Summary:**

Diffuse intrinsic pontine gliomas are malignant brain tumors which arise from the pons in children. These tumors are incurable and nearly all the patients die within a year after diagnosis. To identify effective therapeutics, the molecular mechanisms of tumorigenesis need be comprehensively understood and advanced mouse DIPG models have to be developed for further therapeutic assessment. Over the past decade, remarkable research progress has been made, leading to several ongoing clinical trials. In this review, we update the molecular findings and summarize innovative mouse models generated in the past few years, that are used to understand DIPG and help identify potential treatments. We also prospect future directions for the development of next generation DIPG mouse models.

**Abstract:**

Diffuse intrinsic pontine gliomas (DIPGs) account for ~15% of pediatric brain tumors, which invariably present with poor survival regardless of treatment mode. Several seminal studies have revealed that 80% of DIPGs harbor H3K27M mutation coded by *HIST1H3B*, *HIST1H3C* and *H3F3A* genes. The H3K27M mutation has broad effects on gene expression and is considered a tumor driver. Determination of the effects of H3K27M on posttranslational histone modifications and gene regulations in DIPG is critical for identifying effective therapeutic targets. Advanced animal models play critical roles in translating these cutting-edge findings into clinical trial development. Here, we review current molecular research progress associated with DIPG. We also summarize DIPG animal models, highlighting novel genomic engineered mouse models (GEMMs) and innovative humanized DIPG mouse models. These models will pave the way towards personalized precision medicine for the treatment of DIPGs.

## 1. Introduction

Diffuse intrinsic pontine glioma (DIPG), a high-grade glioma that arises in the pons is predominantly seen in children. This central nervous system (CNS) malignancy represents the leading cause of brain-tumor related death [1]. Ionizing radiation extends overall survival to a median of 11 months [2]. Numerous clinical trials have failed to identify effective agents, or therapeutic combinations against DIPG [3]. A key to identifying improved treatments is to enhance our biological understanding of this tumor.

DIPG molecular signature has been profiled over the last decade. Landmark studies from multiple groups identified an epigenetic oncogenic histone H3K27M mutation in ~80% of DIPGs [4,5,6]. This mutation was defined as a new entity labeled “diffuse midline glioma H3K27M-mutant” in the 2016 World Health Organiztion (WHO) tumor classification [7]. A number of molecular aberrations, which are potential targets for treatment have been identified. They include RB phosphorylation (~30%) [8], p53, Wee1, STAT3, PPM1D (9–23%) and EGFRvIII overexpression [9,10,11,12,13], platelet-derived growth factor receptor A (*PDGFRA*) (30%) and *MET* (26%) amplification [8,14], and *ACVR1* (21%) [15,16,17,18], *PI3K-mTOR* or *-MAPK* (62%) activation [8]. These aberrations are either independent or concurrent with H3K27M mutations in DIPGs. More recently, multiple pathways, such as Notch signaling [19], glycolysis and tricarboxylic acid (TCA) metabolic pathways [20], Wilms’ tumor protein (WT1) overexpression [21,22] and FGFR2-VPS35 fusion [23] were delineated. These pathways are also potential therapeutic targets. Altogether, these abnormalities indicate the molecular complexity of DIPGs. Thus, to identify effective targeting therapeutics or combinations of such, reliable and personalized animal models are desirable for precise preclinical evaluation prior to clinical trials.

Various animal models have been developed for the identification of potential therapeutics against DIPGs. One reason for this development was that historically, surgical biopsies for DIPGs were not performed due to the critical nature of the brainstem, limitations of surgical techniques and histological heterogeneity within the tumor [24]. Multiple murine gliomas induced with various carcinogens were transplanted, so called “allografts”, into the brainstem as models for DIPGs [25]. The results from these models, however, failed to translate into clinical trials due to differences in tumorigenesis mechanisms between these murine gliomas and DIPGs in children. Recently, several groups have safely performed biopsies and obtained tumor specimens from DIPG patients [26]. Moreover, adequate autopsy DIPG specimens have been obtained for experimental purposes. With availability of these invaluable specimens, more accurate patient-derived orthotopic xenograft (PDOX) DIPG mouse models were and continue to be generated for pre-clinical therapeutic testing [27,28,29]. More importantly, these specimens helped generate multiple cell-based models, which have provided significant insights into genetic and epigenetic alterations driving DIPGs. These insights, in conjunction with current cutting-edge gene editing techniques, have allowed and prompted scientists to generate robust genetically engineered mouse models (GEMMs), to understand DIPG tumorigenesis and to provide precise molecular models for pre-clinical testing. One caveat is that the majority of xenograft models are produced in immunodeficient athymic or NOD-SCID gamma (NSG) mice, which lack normal immunity. Consequently, these mouse models do not mimic the human tumor microenvironment including infiltration by immune cells. Recent work has shown that immune cells discovered in pediatric DIPG tumor specimens are distinct and differ from those in adult glioblastomas [30]. Generation of humanized DIPG mouse models is urgently needed to explore these differences, an understanding of which will accelerate therapeutic discovery.

In this review, we summarize current research on DIPG molecular profiling and DIPG animal models, with emphasis on innovative GEMMs and humanized DIPG mouse models. We also discuss the potential of advanced personalized humanized models for pre-clinical therapeutic evaluation, which will pave the way towards personalized precision medicine for DIPG treatment.

## 2. Molecular Characteristics of DIPG

Diffuse intrinsic pontine glioma (DIPG) was identified in 1926 [31]. For decades, its biological behavior was thought to be similar to adult malignant gliomas, thus therapeutic regimens for adult tumors were copied in children. These treatments failed to improve patient outcomes [3], which led researchers to ask if pediatric malignant gliomas including DIPGs fundamentally differ from adult gliomas.

Recently several groups have developed safe and feasible methods to collect biopsy tissue samples from pediatric DIPG patients with minimum mortality [26,32], though routine DIPG biopsy continues to be debated. Furthermore, autopsy samples can be acquired for either direct molecular profiling or development of patient-derived primary cell lines and/or xenografts to gain insight into tumor driving mechanisms [33]. These tissues sources, as well as high-throughput genetic technologies enabled researchers to acquire data, which provided insight into DIPG molecular signatures. For instance, recurrent amplifications of *PDGFRA*, *MET* and retinoblastoma protein (*RB*) are unique for pediatric DIPG [8,34]. These findings prove that DIPG is molecularly distinct from adult malignant gliomas. Other groups using high-throughput genetic sequencing technologies have identified that ~80% DIPGs contain somatic point heterozygous H3.1K27M or H3.3K27M mutations [4,5,6], recognized to be major oncogenic drivers in these tumors [6,35,36].

DIPGs with H3.1K27M or H3.3K27M mutations, coded by *HIST1H3B, HIST1H3C* and *H3F3A*, respectively [37] have different clinical manifestations and gene expression profiles. H3.1K27M tumors are usually restricted to the pons, and show a mesenchymal-like phenotype, with an overall survival of 15 months. Tumors with H3.3K27 mutation are found in the pons and other midline locations such as the thalamus, with an overall survival of 9 months [38,39,40]. These tumors display an oligodendroglial-like phenotype and are more resistant to radiation therapy [40]. Both H3.1K27M and H3.3K27M mutations primarily influence the epigenome and are required for DIPG tumorigenesis and maintenance [41,42]. These mutations predominantly reduce genome-wide levels of repressive H3K27me3 [37,43,44] through inactivation of Polycomb Repressive Complex 2 (PRC2) [45]. Further studies have confirmed that H3K27M suppresses PRC2 through tight binding to EZH2, a core subunit of PRC2 [46]. H3K27me3 levels are differentially associated with H3.1K27M or H3.3K27M mutations, with H3.3K27M epigenome maintaining a certain amount of H3K27me3, while H3.1K27M almost completely excludes genomic-wide H3K27me3 [47]. Moreover, H3K27M DIPG shows slightly elevated H3K27ac [45,48,49], which colocalizes with H3K27M mutations at enhancer or promoter areas [45]. Multiple pre-clinical therapeutic evaluations have shown that inhibition of histone deacetylases is effective and has survival benefits in DIPG xenograft animal models [50], indicating therapeutic potential targeting H3K27ac in H3K27M mutant DIPGs. Interestingly, global H3K4me3 levels are relatively stable regardless of histone H3 mutation [42], however, promoter H3K4me3 level is higher in H3K27M mutants than in WT tumors, at specific gene loci including *Lin28b*, a marker for neural stem cells [51]. More recently, H3K36me2 and H4K16ac were identified as important histone marks in DIPGs [52]. These findings indicated complicated crosstalk among posttranslational histone modifications, the underlining mechanisms are still under investigation.

In addition to epigenomic alterations in H3K27M mutant DIPG, numerous aberrations of gene expression, DNA copy number variations and signal pathways have been uncovered. These may occur concurrently or independently with H3K27M mutation. p38 MAPK is activated in both H3.3K27M and H3.1K27M cultured cells with H3.1K27M tumor cells more sensitive to p38 MAPK inhibition [49]. WNT [49,53], mTOR [53,54] and RTK-RAS-PI3K signaling pathways [53,55] were also active in both tumor subtypes. Interestingly, H3.1K27M and H3.3K27M DIPGs have their own associated mutations as summarized in Figure 1.

The diversity in genetics, chromatin landscape and metabolic reprogramming of DIPGs clearly shows that individualized therapeutics will be critical for effective treatments. To this end, developing personalized animal models for pre-clinical assessment is an important step to identify and determine the best therapeutic agents.

## 3. Murine Models of DIPG

Animal models for brain tumors are critical for understanding potential tumorigenesis mechanism, and to assess the efficacy of novel therapeutic molecules or compounds for clinical application. Several species including zebrafish [56], rats [57,58], mice [59,60,61], dogs [62], hamsters [63], monkeys [64] and non-human primates [65] have been used to develop brain tumor models. Mice are the most popular animal because they are easy to handle and manipulate and have a short lifespan. Thus, here we focus on summarizing mouse brainstem tumor models that expand biological understanding of DIPG.

### 3.1. Syngeneic Brainstem Glioma Models

The first animal models for brainstem glioma were developed in rats at 2002. Rat glioma F98 and 9 L cells were implanted using a stereotactic head frame [59]. Following this study, multiple rat syngeneic cell lines including C6, F98, 9 L with or without luciferase modification were inoculated into the brainstem of neonatal or adult rats to generate brainstem tumor models [25,57,66,67,68]. These models were developed using stereotactic head frames targeting the pons (Figure 2A). Experimental rats developed pontine tumors, with appropriate location and microenvironment. However, most of these cell lines were developed from cerebral cortical tumors and heavily passaged in culture [58], thus they were not the best model to represent biological behavior of childhood DIPGs. These models are now being replaced by xenograft models using human DIPG cells directly from biopsy/autopsy or primary culture [69].

### 3.2. Patient-Derived Orthotopic Xenograft (PDOX) DIPG Mouse Models

Human glioblastoma (GBM) xenografts were developed by Hashizume et al. in 2010 in mouse or rat brainstem as models for DIPG [60,70]. Adult GBM cells including U87, U251MG and GS2 were inoculated into the pons using a stereotactic headframe to develop brainstem tumors for therapeutic testing. In these studies, brainstem microenvironment and blood-brain barrier were taken into consideration, however, as most of these cell lines were derived from adult cerebral cortical GBM, biological features of DIPG were ignored. A few studies with ionizing irradiation, which is standard care for pediatric DIPG, were tested using these models and showed temporary efficacy [28]. However, other therapeutic tests did not show efficacy in multiple clinical trials. For instance, temozolomide, a DNA methylation agent for adult GBM treatment, and PD-0332991, a CDK4/6 inhibitor were reported effective in murine brainstem models using adult GBM cell lines [60,61,71], however, clinical trials using TMZ and CDK4/6 inhibitor did not markedly extend survival for children with DIPG [72,73]. These findings indicate that glioma cells from adult tissue and from non-pontine locations are not suitable for predicting DIPG therapeutic response.

Due to limitations of the models discussed above, human DIPG cells obtained from autopsy and biopsy samples were developed by Monje et al. in 2011. These cell sources provided new biological insights into DIPG molecular characteristics and were key in moving forward the development of patient-derived orthotopic xenograft (PDOX) DIPG models. The first pediatric DIPG in vitro cell culture and in vivo xenograft mouse models were developed using early postmortem tumor tissue [74], in which pontine histopathology was similar to that of DIPG. Following this work, primary cultured cells from DIPG biopsy specimens and an orthotopic mouse model using these cells were successfully developed by Hashizume et al. in 2012 [70]. These pioneering studies led to the development of numerous primary cultured DIPG cells and PDOX models for testing novel therapeutics (Figure 2B). A DIPG model in NOD-SCID mice was created by Harutyunyan et al. in 2019 using CRISPR-Cas9 gene editing to manipulate H3K27M expression in patient derived primary culture DIPG cells. This model was used to observe molecular alterations and tumorigenesis in the absence and presence of the H3K27M mutation [41]. Recent studies have shown that promoter H3K27Ac is elevated in DIPG with H3K27M mutation [45,48]. Several PDOX models for DIPG were used to test panobinostat, a histone acetylation inhibitor, which showed promising results [48,75]. Excitingly, panobinostat has been registered in multiple clinical trials for DIPG (NCT02717455, NCT04341311, NCT03566199). In addition to panobinostat, more compounds and novel small molecules have been tested in various DIPG PDOX models as summarized in Table 1.

Overall, stereotactic biopsy of DIPG is considered safe and effective [94,95], and techniques to develop primary cell cultures and orthotopic DIPG models from autopsy specimens have improved. This has led to the development of PDOX mouse models with primary cultured DIPG cells. These models will help identify effective therapeutic pharmaceutical agents or compounds for clinical trials.

### 3.3. Genetically Engineered Mouse Models (GEMM) for DIPG

Xenograft models cannot answer some fundamental questions, for instance, (i) what cells does DIPG originate from? (ii) is a specific genetic or epigenetic alteration sufficient to drive DIPG formation? and (iii) what role do oncogenes or tumor suppressors play in DIPG tumorigenesis? Genetically engineered mouse models (GEMMs) can be used to help answer these critical questions.

GEMMs are ideal to investigate DIPG cell origin for insight into tumorigenesis mechanisms. Several systems, including replication-competent avian sarcoma-leukosis virus long terminal repeat splice acceptor (RCAS/Tv-a) (Figure 3A), Sleeping Beauty/PiggyBac transposon in combination with lentivirus (Figure 3B) and Nestin-Cre/LoxP recombination (Figure 3C) systems have been utilized to generate multiple GEMMs. GEMMs developed with these systems can be used to investigate DIPG tumorigenesis [41] and potential populations of cells from which DIPG can originate including: nestin-expressing progenitor cells (nestin^+^/vimentin^+^/Olig2^+^) [74]; oligodendrocyte progenitor cells (Olig2^+^/ Sox2^+^/APC^−^) [96]; and at least two distinct types of Pax3^−^ expressing progenitor cells (immature Pax3^+^/Nestin^+^/Sox2^+^ progenitor and differentiated Pax3^+^/NeuN^+^ neuron) [97]. In addition to these GEMMs, nestin positive progenitor cells which line the floor of the fourth ventricle genetically engineered with ectopic *PDGFB*, *TP53* loss and with or without the H3.3K27M mutation with the RCAS/Tv-a system (Figure 3A), were also shown to have biological similarity with DIPG [37,98,99]. Given the fact that the majority of human DIPGs are believed to arise from the ventral pons, and its tumorigenesis is complicated and regulated by dynamic tempo-spatial genomic and epigenomic events, cell origin for DIPG is a key question which is yet to be determined.

GEMMs are robust tools to identify tumor drivers. Somatic point heterozygous mutation on *HIST1H3B* or *H3F3A* and multiple deregulation of genes such as *TP53*, *PDGFB*, *PDGFRA*, and *ACVR1* are important genomic alterations. While *HIST1H3B* or *H3F3A* which codes for the H3K27M mutation was initially considered a tumor driver, human embryonic stem cell derived neural progenitor cells (hES-NPCs) transformed using a combination of lentivirus encoding H3.3K27M did not form tumors in mouse pons. hES-NPCs engineered with H3K27M, *TP53* loss and *PDGFB* mutation formed tumors in mouse pons, with biological characteristics similar to DIPG [100]. Another GEMM created via in utero electroporation to deliver Piggy/Bac transposable-H3.3K27M, *TP53* CRISPR/Cas9 and *PDGFRA* into the lower rhombic lip of NPCs in vivo (Figure 3B), formed brainstem tumors with biological characteristics similar to DIPG [101]. Recently, a novel inducible GEMM was developed in which H3.3K27M was transduced into an *H3F3A* locus in combination with loss of *TP53* and a *PDGFRA* mutant. This combination was driven by a tamoxifen-inducible Cre recombinase in neonatal nestin-positive cells throughout the developing brain (Figure 3C) and formed spontaneous malignant brain tumors which mimic H3.3K27M DIPG [51]. Using a similar approach, H3.1K27M and *ACVR1^G328V^* were knocked into their respective loci driven by Cre recombinase in Oligo2^+^ oligodendrocyte precursor cells (OPCs). These genetically engineered cells showed glial differentiation arrest and high proliferation but, were insufficient to drive tumor formation. However, spontaneous high-grade gliomas were formed in the brainstem and thalamus if additional endogenous *PIK3CA^H1047R^* was knocked in [15]. These GEMMs indicate that the oncohistone mutation of H3.1K27M or H3.3K27M in combination with *PDGFRA*, *TP53* loss, *PDGFB*, *ACVR1* mutation have a synergistic effect on driving DIPG.

Numerous dynamic genomic and epigenomic alterations contribute to DIPG tumorigenesis. GEMMs can be used to identify therapeutics that target for specific gene alterations or combinations of changes. For instance, a high-grade brainstem glioma was generated by overexpression of *PDGFB* in combination with *Ink4a-ARF* loss in the posterior fossa of neonatal Nestin Tv-a mice. This model was used to test perifosine, an AKT inhibitor and irradiation [99]. The model was also used to test PD-0332991 (PD), a CDK4/6 inhibitor, which did prolong survival [98]. Another GEMM for DIPG driven by *PDGFB*, H3.3K27M, and *TP53* loss using the RCAS/tv-a system was developed to test BMS-754807, a potent and reversible small molecule multi-kinase inhibitor, which showed significant efficacy in vitro [102]. A GEMM model of hES-NPCs engineered with H3K27M, *TP53* loss and *PDGFB* mutation was used to test menin inhibitor MI-2, which showed significant efficacy in decreasing tumor growth [101].

In all, GEMMs for DIPG are an important supplement to PDOX for investigation of tumorigenesis mechanisms in an immunocompetent microenvironment, to determine cell origin and to test novel therapeutic compounds. However, GEMMs generally focus on a few genes, only represent tumors with engineered backgrounds and lack heterogeneity of patient tumor samples. Due to the differential tumor associated microenvironment between mice and humans, tumor biological behavior may differ from original human DIPGs. Thus, precise DIPG animal models which incorporate human immunity must be developed.

### 3.4. Fidelity of PDOX and Recapitulation GEMMs of Human DIPG

Mouse models that faithfully maintain molecular characteristics of original tumors are critical to evaluate therapeutic efficacy of novel therapies. To this end, models with high biological and molecular fidelity and recapitulation of human DIPG are also important and essential.

PDOX DIPG models directly transfer fresh biopsy or autopsy tumor tissue into immunodeficient mouse brains without manual manipulation, for instance, cell culture, transfection or transduction etc. Their fidelity has been comprehensively analyzed with cutting-edge high throughput sequencing techniques, and the results show that these models faithfully recapitulate the molecular signature of initial derived pediatric DIPG [103,104,105,106,107]. Results from a recent report support and further emphasize that PDOX DIPG models recapitulate parental tumor tissues, through robust and integrative analysis with histopathology, DNA methylation, exome and RNA-sequencing [108]. Primary cultured DIPG cells are also important sources for indirect PDOX DIPG models [109], which retain biological and molecular characteristics of original derived tissues (Table 2). For example, PDOX DIPG models, PED17 created with both fresh tissue and primary cultured cells in Mayo Clinic, which faithfully recapitulate patient derived tumor characteristics through multiple passages [109]. This finding is supported by recent analysis from a large panel of DIPG cells [108].

GEMM models are generally used to investigate DIPG cell origin for insight into tumorigenesis mechanisms. “Hot” spot mutation on *HIST1H3B* or *H3F3A* in combination with deregulation of multiple genes including *TP53* loss, PDGFB and PDGFRA overexpression, *ACVR1* mutations are genetically engineered to create pediatric DIPG models as described in Figure 3. The fidelity of these models was compared in several studies (Table 2). GEMM with H3.3K27M^+^; Pax3^+^; p53^−^; PDGFB^+^ has the highest molecular and biological fidelity of pediatric DIPG, in comparison to either Pax3^+^; PDGFB^+^ or Pax3^+^; p53^−^; PDGFB^+^ [97]. Models created with engineered *ACVR^R206H/G328V^*; H3.1K27M; p53^−^; PDGFA^+^ more faithfully recapitulate pediatric DIPG than *ACVR^R206H/G328V^*; H3.1K27M and *ACVR^R206H/G328V^*; H3.1K27M; p53^−^ [17]. The fidelity of the models engineered with H3.3^K27M^; PDGFA^+^; Trp53^−^ is higher than the ones engineered with H3.3^WT^; PDGFA^WT^; Trp53^−^ [101].

PDOX and GEMM are two different categories of DIPG models with differential fidelity. In general, PDOX models created with fresh tumor tissues have the highest biological and molecular fidelity, in comparison to the PDOX models with cultured DIPG cells and GEMM DIPG models.

## 4. Humanized Mouse Models for DIPG Precision Medicine

Recently, serval studies have shown that the tumor immune microenvironment (TIME) has critical roles in DIPG: (i) Tumor-infiltrating cells (TILs) including Treg, CD4 T cells, NK, B cells, monocytes and eosinophils have been identified in H3K27M mutant DIPGs [110]; (ii) Expression of indoleamine 2,3 dioxygenase 1 (IDO1), an immunosuppressive enzyme that metabolizes tryptophan, is low in cultured DIPG cells. However, in vitro induction of IDO1 with IFNγ showed potential therapeutic value [111]; (iii) the disialoganglioside GD2 is highly expressed in patient-derived H3K27M mutant glioma cell cultures. Anti-GD2 CAR T cells incorporating a 4-1BBz costimulatory domain demonstrated robust antigen-dependent cytokine generation and killed DIPG cells in vitro [112] and (iv) a humanized anti-CD47 antibody, Hu5F9-G4, demonstrated therapeutic efficacy [113].

These cutting-edge findings require testing using reliable animal models prior to translation into clinical trials. For many years, chimpanzees were used to bridge the gap between rodent models and humans. However, the biomedical use of chimpanzees is prohibited in Europe and the United States. Therefore, to overcome the limitations of translating laboratory rodent discoveries into clinical applications, development of mouse models that closely recapitulate human biological systems, labeled as “humanized” mice is critical for pre-clinical investigation. In this section, we will discuss recent progress in the development and potential use of humanized DIPG mouse models.

Humanized mice are defined as immunodeficient mice engrafted with human tissues [114], which include PDX or PDOX models. Discovery of nude mice in the 1960s and severe combined immunodeficient (SCID) mice in the 1980s [115] were key advances for xenografts. Following these models, non-obese diabetic (NOD)/SCID and NOD/SCID /β2m^null^ and NOD/Rag1^null^Pfp^null^ mice were subsequently developed from NOD/SCID and NOD/Rag1^null^mice [115], which contribute to humanized mouse generation. Another landmark advance in the generation of humanized mice was the generation of NOD/SCID/γc^null^ mice and Rag1/2^null^γc^null^ mice through introducing *IL2ra* into NOD/SCID and RAG1/2^−/−^ mice in the 2000s [115]. These mice show multiple immunodeficiencies including impaired T, B and nature killer (NK) cells, and reduced macrophages and dendritic cell immune function, which show a high rate of human cell including DIPG engraftment. More recently, several humanized mouse models were used to test novel potential therapeutics for DIPG. NRG (NOD.Cg-*Rag1^tm1Mom^ Il2rg^tm1Wjl/SzJ^*) mice, which had DIPG cells implanted in the pons, were used to test therapeutic efficacy of a DNA-damaging reagent 6-thio-2′deoxyguanosine (6-thio-dG). The results demonstrated promising therapeutic efficacy [86]. NOD-SCIDγ (NSG) mice were used to test anti-CD47 antibody Hu5F9-G4 through orthotopic injection of several DIPG cells. The results showed intraperitoneal treatment with this antibody significantly reduced tumor growth and showed significant survival benefit [113]. NSG NOD-SCID *IL2rg*^−/−^ (NSG) mice were used for testing anti-GD2 CAR-T cell immunotherapy [112], with the results leading to an active clinical trial (NCT04196413).

Humanized mice are also defined as immunodeficient mice engrafted with hemato-poietic cells [114,115]. There are several strategies for establishment of these humanized mouse models (Figure 4). The first one is humanized mice receiving human peripheral blood mononuclear cells (PBMCs) engrafted to establish the Hu-PBL-SCID model. This model is suitable for short-term research and investigation of the relationship between immune function of lymphocytes in peripheral blood and tumor biological behavior [116]. The second strategy is the transfer of human hematopoietic stem cells (HSCs) into mice with the *IL-2rγ^null^* mutation to develop the Hu-SRC-SCID model [117]. HSCs can be obtained from granulocyte colony-stimulating factor-mobilized PBMCs, adult bone marrow, fetal liver and umbilical cord [118]. This model also supports engraftment of complete human immune system through injection of CD34^+^ HSCs (Hu-CD34^+^ model), which is appropriate for investigating tumor growth and immune system development [119]. The third model is a bone marrow/liver/thymus (BLT) model which is developed via transplanting human fetal liver and thymus under the kidney capsule. More recently, a novel and revolutionary humanized mouse model NOD-SCID *IL2rg^null^* SCF/GM-CSF/IL3 strain engrafted with human thymus, liver, and hematopoietic stem cells (termed Bone marrow, Liver, Thymus [BLT]) (NSG-SGM3-BLT) was used to develop an orthotopic model through injection of SF8628, a H3.3K27M mutant cell in the pons, to test IDO1 induction by CD4^+^ and CD8^+^ T cells. The results showed that these T cells directly increase IDO1 expression in intracranial DIPG tumor and are thus a promising adjuvant immunotherapy [111].

In addition, mice transgenically engineered to express human genes are also humanized mice. One example is MISTRG mice, in which seven genes including *M-CSF^h/h^ IL-3/GM-CSF^h/h^ SIRPa^h/h^ TPO^h/h^ RAG2^−/−^ IL2Rg^−/−^* were knocked into mouse genomic loci [120]. MISTRG mice are humanized with high immunodeficiency, which prevents immune rejection of the human grafts. These mice are a robust tool to investigate engrafted tumors and innate immune response, which is potentially useful for the development of humanized DIPG models.

Altogether, these advanced humanized mouse models provide a more realistic human tumor immune microenvironment with potential for better drug response and prediction for clinical trials and will help to identify effective therapeutic regimens for DIPG.

## 5. Summary and Perspectives

In summary, we review up-to-date genomic and epigenomic profiling advances in DIPG. In addition to H3K27M mutation, numerous aberrations of gene expression, DNA copy number variations, abnormal signal pathways and posttranslational histone modifications also contribute to DIPG tumor biology. This highlights the complexity and challenges of identifying effective therapeutics for individual patients. We also summarize mouse models for DIPG with a specific highlight on GEMMs and recent innovative humanized mouse models. Currently, neither therapeutic strategies nor compounds developed using DIPG mouse models have been approved for the treatment of DIPG. However, several promising clinical trials are in progress. We believe that because of rapid advances in biological techniques such as high-throughput proteomics analysis, next-generation and single cell sequencing, advanced gene-editing tools etc., more novel molecules or signal pathways and epigenetic factors as potential therapeutic targets will be identified. Testing these targets in novel GEMMs and innovative humanized mouse models, will pave the way to precision medicine necessary for effective treatments, that have the potential to improve outcomes for children with these tumors.

## Figures and Tables

**Figure 1 cancers-13-01114-f001:**
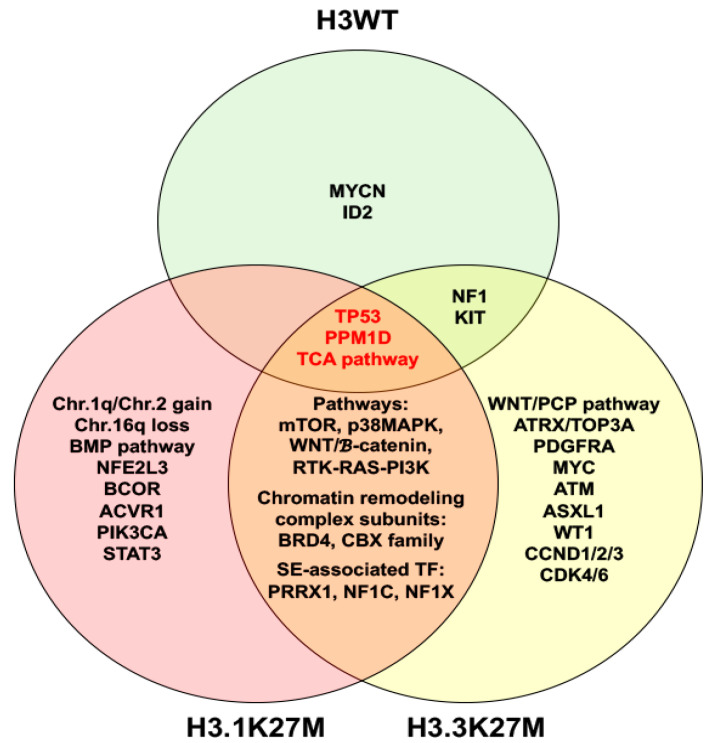
Venn diagram illustrates molecular characteristics of histone H3 wild type (H3WT), H3.1K27M and H3.3K27M diffuse intrinsic pontine gliomas. Abbreviation: TCA, tricarboxylic acid.

**Figure 2 cancers-13-01114-f002:**
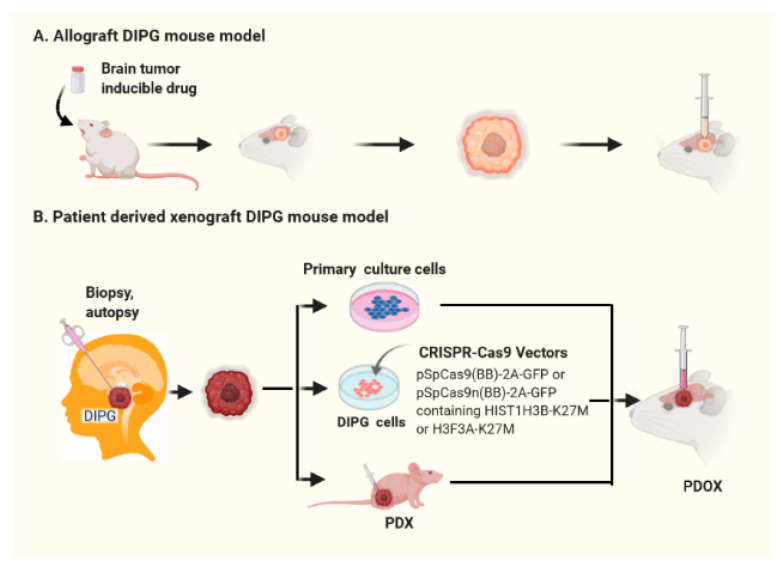
Schematic diagrams illustrate the development mouse models for DIPG. (**A**) Allograft mouse models. (**B**) Patient derived xenograft (PDX) or patient derived orthotopic xenograft (PDOX) mouse models using primary culture cells with or without modification using CRISPR-Cas9 gene editing, and viable tumor biopsy or autopsy specimens.

**Figure 3 cancers-13-01114-f003:**
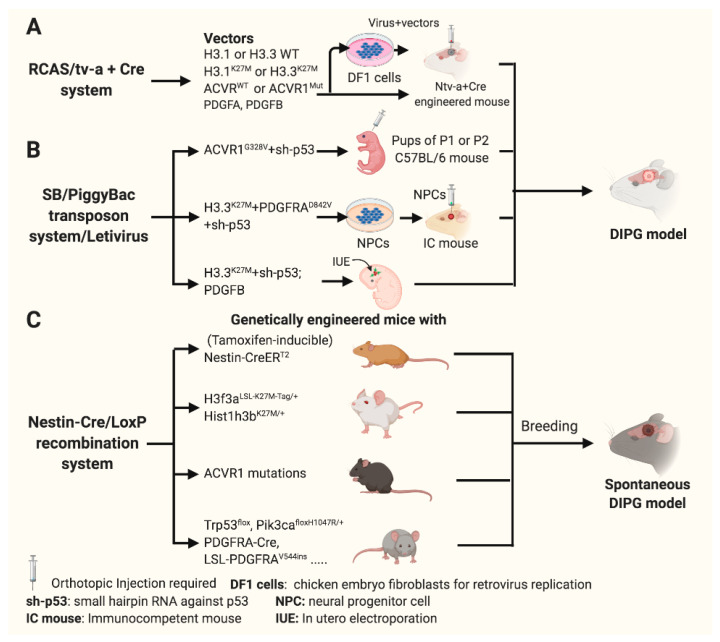
Genetically engineered mouse models (GEMMs) for DIPG using RACS/tv-a, SB/PiggyBac transposon with lentivirus, and Nestin-Cre/LoxP recombination systems. Schematic illustration demonstrating generation of GEMMs for DIPG. (**A**) Vectors were constructed with replication-competent avian sarcoma-leukosis virus long terminal repeat splice acceptor (RCAS/Tv-a). These viral vectors were either amplified with DF1 cells prior to inoculation ((**A**) top panel) or directly inoculated ((**A**) bottom panel) into the pons of Nestin-Cre engineered Ntv-a or Ntv-a; Ink4a-ARF^−/−^; Ntv-a; p53^fl/fl^, and Ntv-a; PTEN^fl/fl^ mice. (**B**) Vectors constructed with the Sleeping Beauty Piggy/Bac transposon in combination with lentivirus system were injected into postnatal day 1 (P1) or 2 (P2) C57BL/6 mice ((**B**) top panel) or transduced into neural progenitor cells (NPCs) ((**B**) middle panel) followed by orthotopic NPC pontine inoculation, vectors were also administered via in utero electroporation (IUE) ((**B**) bottom panel). (**C**) Mouse models with various molecular backgrounds associated with DIPG were developed with the Nestin-Cre/LoxP recombination system and bred to generate GEMMs for spontaneous DIPG.

**Figure 4 cancers-13-01114-f004:**
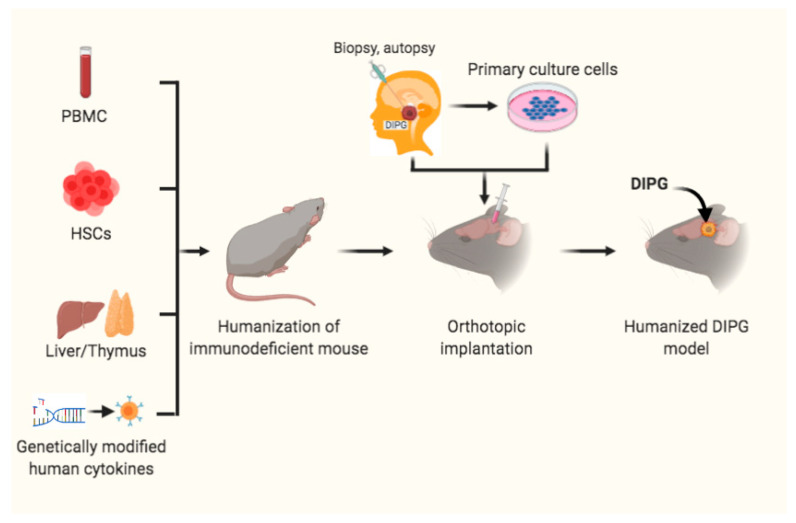
Humanized mouse models for DIPG. Immunodeficient mice received peripheral blood mononuclear cells (PBMCs), or hematopoietic stem cells (HSCs) and transplantation of liver/thymus, or genetically modified human cytokines prior to orthotopic intracranial inoculation of DIPG cells, from patient biopsy or autopsy specimens or primary cultured cells, into the pons to develop humanized DIPG models.

**Table 1 cancers-13-01114-t001:** Patient derived DIPG cell lines for in vitro culture and for in vivo generation of orthotopic mouse models.

Cell Source	H3 Mutation	Name of Cell Line	Novel Therapeutic Compounds Tested
Autopsy	H3WT	VUMC-DIPG-10 [76], DIPGM(T) [74,77]	OTSSP167 [76], veliparib, olaparib, niraparib [77]
	H3.1K27M	SU-DIPG-IV [17,45,77,78,79,80,81,82]	LDN212854 [17], JQ1 [45], Panobinostat, GSK-J4 [81], veliparib, olaparib, niraparib [77], Corin [80], PTC-209 [82]
	H3.3K27M	JHH-DIPG1 (T) [19,54,76,81,83,84]	Delta-24-RGD [83], TMZ [84] *, GSK-J4 [81], Panobinostat, OTSSP167 [76], TAK228 [54], MRK003 [19]
		SU-DIPG-VI (T) [85], XIII, XVII [19,54,78,79,80,81,82,86,87]	LDN-193189, LDN-214117, LDN-212854 [79], TAK228 [54], 6-thio-Dg [86], BGB324 [87], Panobinostat [81], GSK-J4 [81], Corin [80], MRK003 [19], PTC-209 [82]
Biopsy	H3WT	CCHMC-DIPG-1 (T) [82,86]	PTC-209 [82], 6-thio-Dg [86]
	H3.1K27M	HSJD-DIPG-018 [79]	GSK343 [88], EPZ6438 [88]
		VUMC-DIPG-B [81]	Panobinostat, GSK-J4 [81]
	H3.3K27M	SF8628 (T) [13,17,45,89], SF7761(T) [19,45,54,84,87,90,91]	TAK228 [54], MK-1775 [13], JQ1 [45], TMZ [84], Panobinostat, GSK-J4 [81,90], BGB324 [87], CUDC-907 [89], MRK003 [19], GSK343 [88], EPZ6438 [88]
		HSJD-DIPG-007 (T) [79], 008, 012, 014 [78,79,80,87,90,92,93]	Bevacizumab [93], OTSSP167 [76], BGB324 [87], Panobinostat [92], LDN-193189, LDN-214117, LDN-212854 [79], GSK343, EPZ6438 [88], Corin [80]
		VUMC-DIPG-A (T) [81,87] [76], F(T) [88,91]	OTSSP167 [76], BGB324 [87], Panobinostat [81]
		TP54, 80 (T) [83], TP83, 84 [83]	Delta-24-RGD [83]
		NEM 157, 163, 165, 168 [81]	Panobinostat, GSK-J4 [81], Delta-24-RGD [83] *
		QCTB-R059(T) [79,91], CHRU-TC68 [16]	LDN-193189, LDN-214117, LDN-212854 [16,79]
		CCHMC-DIPG-2 [82]	PTC-209 [82]

Note. T: tumorigenic; OTSSP167: MELK inhibitor; LDN212854, LDN-193189, LDN-214117: BMP receptor inhibitor; GSK-J4: KDM6B-specific inhibitor; PTC-209: BMI-1 inhibitor; TMZ: Temozolomide; TAK228: oral dual TORC1/2 inhibitor; MRK003: γ-secretase inhibitor; BGB324: Bemcentinib; GSK343: EZH2 inhibitor; EPZ6438: Tazemetostat; MK-1775: Adavosertib; JQ1: BET bromodomain inhibitor; CUDC-907: dual PI3K and HDAC inhibitor; * clinical trial.

**Table 2 cancers-13-01114-t002:** Summary of PDOX and GEMM models’ fidelity and recapitulation of human DIPG, analyzed with histology/pathology and/or high-throughput characterization.

Model	Cell Resource or Genetic Engineering	Histology	Molecular Analysis	Fidelity	Reference
PDOX	Biopsy: DIPG tissue, PED17	HE; IHC	RNA-seq, WGS, WES-seq DNA methylation-seq	High	[103,105,108]
	Autopsy: DIPG tissue				[73]
	Cultured DIPG cells, PED17				[108,109]
GEMM	H3.3K27M + *TP53* loss +*PDGFRA* activation	IF	RNA- & ChIP-seq	High	[99]
	Pax3^+^; PDGFB^+^ Pax3^+^; p53^−^; PDGFB^+^ H3.3K27M^+^; Pax3^+^; p53^−^; PDGFB^+^	IHC; IF	N/A	Low Moderate High	[96]
	*ACVR^R206H^*; H3.1K27M *ACVR^R206H/G328V^*; H3.1K27M; p53^−^ *ACVR^R206H/G328V^*; H3.1K27M; p53^−^; PDGFA^+^	IHC	RNA-seq	Low Moderate High	[17]
	*ACVR^G328V^*; *PIK3CA^H1047^*; Oligo2^+^ *ACVR^G328V^*; *HIST1H3B^K27M^*; *PIK3CA^H1047^*; Oligo2^+^	IHC	RNA-seq	Low High	[15]
	H3.3WT; PDGFRA; p53^cKO^ H3.3K27M; PDGFRA; p53^cKO^	IHC	RNA- & ChIP-seq	Low High	[101]
	H3.3^WT^; PDGFA^WT^; Trp53^−^ H3.3^K27M^; PDGFA^+^; Trp53^−^	IHC	RNA- & ChIP-seq, WES	Low High	[110]

Abbreviations: HE: hematoxylin and eosin; IHC: immunohistochemistry; WGS: whole-genome sequence; WES: whole-exome sequence; ChIP: chromatin immunoprecipitation; KO: knockout; N/A: not applicable.

## Data Availability

Data sharing is not applicable to this article as no new data were created or analyzed in this study.

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
