# Peer review of "Advanced Pediatric Diffuse Pontine Glioma Murine Models Pave the Way towards Precision Medicine"

_cancers, 2021, doi:10.3390/cancers13051114_

Round 1
Reviewer 1 Report
This review from Chen and colleagues is a nice and detailed summary of the ranges of models available and in development for the study of DIPG. The authors provide good detail on many of the models, and focus on the utility of the models for particular studies including genetic studies to determine the roles for specific genomic features of DIPG. This is a strength of the review. There are some minor issues I would have with the emphasis placed on some of the various models and this could be considered in a revised version of the manuscript. I have listed these points below
- The early models of rat glioma cell lines and human glioma samples are largely now redundant on the field. There is perhaps too much space devoted to these. As a minor point, these are referred to as "synergetic allograft" models but I suspect that the authors mean syngeneic? The term allograft can be used alone.
- More discussion and emphasis might be given to models such as Harutyunyan (Nature Comms 2019) where the role of the H3K27M mutation was investigated using CRISPR modifications. This paper was referenced but this model deserves more emphasis.
- One feature that was limited was a discussion of how the PDOX models retain genetic fidelity (do they retain or lose genomic features present in primary tumours) and how well the GEMMs mimic the broader molecular and phenotypic features of the human disease. A table to address this might be a valuable inclusion
Reviewer 2 Report
This is a well-written review that will be useful to the field. Several minor comments:
- It would be useful to mention the % prevalence of each mutation listed at the top of page 2 and bottom of page 3
- Section 3, some idea of which years each model categories were first made in would be nice.
- Section 3.3, should add that benefit of GEMMs is also immunocompetence. Additionally, what are the pros and cons of each GEMM type (RCAS vs SB vs Cre) compared to each other? Which ones most closely recapitulate human DIPG?
- Section 4, the first models described, it's not clear how they differ from PDOX, other than use of highly immunodeficient mice. The term "humanized models" is generally used to mean NSG-type mice reconstituted with human immune cells so as to determine their effect on the engrafted tumor. The way the term is used here is confusing.
- EZH2 and BET inhibitors are a hot topic in DIPG research and should be mentioned more prominently, especially their use in mouse DIPG models (refs 44, 47, and 91).
- Do any of the models have additional molecular analyses which may suggest fidelity to human DIPG, such as RNAseq, epigenetic assays (ChIP-seq, ATAC-seq), or proteomics? How lethal/aggressive are the mouse DIPG models compared to human DIPG?
Reviewer 3 Report
This is an interesting and well-written paper. It summarizes nicely the different animal models of pediatric diffuse intrinsic pontine glioma.
Since precision medicine is mentioned in the title, the authors should discuss more the encouraging therapeutic responses obtained in the animal models.
Some sentences/parts of the manuscript are redundant. The introduction could be shortened.
DIPG are not exclusively seen in children as stated at the beginning of the introduction. Such tumors may be diagnosed in adults as well even if rarely.
Minor points:
- 2016 CNS tumor classification (introduction part) : it should be mentioned that it is the 2016 WHO classification. Reference 7 is not the most appropriate here. The 2016 classification published by IARC Lyon (blue book series, Louis et al.) should be cited instead.
- The genes should appear in italics. It is TP53 gene and not p53.
- Second paragraph of part 2 (Molecular characteristics of DIPG) : "longer" and "shorter" survival times. The authors should be more precise (state the difference in survival in weeks).
- Paragraph 3 : "for understanding potential": is a word missing here? "Murine is the most popular": mouse or mice instead?
Reviewer 4 Report
cancers-1108627
Reviewer comments:
In this review, the authors summarized the current research on DIPG molecular profiling and murine model. More specifically, after summarizing the molecular characteristics of DIPG, they discussed the role of different murine models in the difficult task to improve the prognosis of this lethal disease. They detailed the different models pointing out the progress made thanks to each model but also the limitations.
Therefore, they reviewed the synergetic allograft brainstem glioma models, the patient-derived orthotopic xenograft (PDOX) DIPG mouse models and the genetically engineered mouse models (GEMM) for DIPG.
The authors must congratulate for their work. However, I have several concerns regarding the manuscript.
1) The title needs to be change from “Animal models” to “Murine models” as they only reviewed murine models.
2) I thought that the second part “2. Molecular characteristics of DIPG” is too long. Indeed, the principal aim of the review seems to be the role of the murine models in the progress of DIPG treatments. Therefore, the long description of molecular profiling of DIPG seems too much.
3) Even if the authors tried to exposed the limitations of each models, they do not state clearly that, until now, no model has ever provided the slight benefit for the patient. I think that this fact must be put in the limitation and discussed. Indeed, the surgical procedure needed to obtain tissue samples from the patient, is not devoted from complications. Regarding, this specific point, the authors should temper their statement because even if the article they cited (Roujeau et al) reported few complications, a large body of the literature do not report the same experience. Therefore, the current protocols that require surgical biopsies are more and more discussed in the pediatric neurosurgical and oncological community because we are faced with ethical problems when we propose a biopsy to parents knowing that it will prolonged the child hospital stay and probably worsen (even transitorily) the clinical status.
Round 2
Reviewer 4 Report
The authors made some great improvements in the manuscript. However, they did not take into consideration one of my main concern which was that "they do not state clearly that, until now, no model has ever provided the slight benefit for the patient and I think that this fact must be put in the limitation and discussed". Indeed they did not temper their conclusion and still conclude "and dramatically improve outcomes for children with these tumors" (which is absolutely not true until now). I really believe that this must be addressed by the authors before the paper could be accepted.
